# Drug-Induced Esophageal Ulcer in Adolescent Population: Experience at a Single Medical Center in Central Taiwan

**DOI:** 10.3390/medicina57121286

**Published:** 2021-11-23

**Authors:** Shu-Wei Hu, An-Chyi Chen, Shu-Fen Wu

**Affiliations:** 1Department of Pediatrics, Tungs’ Taichung MetroHarbor Hospital, No. 699, Sec. 8, Taiwan Blvd., Wuqi Dist., Taichung City 435403, Taiwan; t13223@ms.sltung.com.tw; 2Division of Pediatric Hepatology and Gastroenterology, China Medical University Children’s Hospital, China Medical University, No. 2, Yude Rd., North Dist., Taichung City 404327, Taiwan; d8427@mail.cmuh.org.tw; 3School of Medicine, College of Medicine, China Medical University, No. 91, Xueshi Rd., North Dist., Taichung City 404328, Taiwan

**Keywords:** adolescent, drug-induced esophagitis, esophageal ulcer, endoscopy, management

## Abstract

*Background and Objectives*: Drug-induced esophageal ulcer is caused by focal drug stimulation. It may occur in adults and children. Limited research is available in pediatric patients with drug-induced esophageal ulcer; therefore, we designed this study to determine the characteristics of this disease in this population. *Materials and Methods*: Thirty-two pediatric patients diagnosed with drug-induced esophageal ulcers from a hospital database of upper gastrointestinal tract endoscopies were included. After treatment, patients were followed for 2 months after upper gastrointestinal endoscopy. *Results*: Female patients were predominant (56.2%/43.8%). The mean age of patients was 15.6 years (median, 16 years; interquartile range, 2 years). Doxycycline was administered in most cases (56.3%); other drugs were dicloxacillin, amoxicillin, clindamycin, L-arginine, and nonsteroidal anti-inflammatory drugs. Doxycycline was associated with kissing ulcers. Esophageal ulcers induced by nonsteroidal anti-inflammatory drugs were more often associated with gastric or duodenal ulcers. The most common location was the middle-third of the esophagus (78.1%). Patients were treated with proton pump inhibitors, sucralfate, or H2-blockers. The mean duration for which symptoms lasted was 9.2 days. No esophageal stricture was found in 24 patients who were followed for 2 months after upper gastrointestinal endoscopy. *Conclusions*: The authors suggest informing patients to take medicine with enough water (approximately 100 mL) and enough time (15–30 min) before recumbency, especially high-risk drugs, such as doxycycline or nonsteroidal anti-inflammatory drugs.

## 1. Introduction

A drug-induced esophageal ulcer results from chemical injury on esophageal mucosa. The incidence is estimated at 3.9 per 100,000 individuals per year [1]. Odynophagia, dysphagia, and chest pain are the usual symptoms of drug-induced esophagitis, but mediastinum perforation, hemorrhage, and death also occur [2]. Many clinicians do not recognize drug-induced esophageal ulcers as a cause of chest pain or odynophagia. Most patients report self-limited symptoms; therefore, this diagnosis is often underestimated [3]. To our knowledge, more than 100 drugs can cause drug-induced esophagitis. The major risk factors are insufficient water or lying down after taking medicine. Most drug-induced esophagitis studies are conducted with adults, and there is scant evidence in the pediatric population [4,5]. However, patients with this problem are still encountered in clinical work.

This study aimed to describe the clinical characteristics of drug-induced esophageal ulcers in children and discuss factors such as age, sex, drugs, ulcer locations, treatment, and outcome.

## 2. Materials and Methods

This study was a retrospective analysis of 32 patients diagnosed with drug-induced esophageal ulcers by upper gastrointestinal endoscopy from September 2011 to April 2021 in a children’s hospital in central Taiwan. This hospital performs upper gastrointestinal endoscopy for approximately 250 pediatric patients every year. During this retrospective period, 2257 upper gastrointestinal endoscopies were performed by pediatric gastroenterologists, of which 70 patients with esophageal ulcers were selected. We selected patients with drug-induced esophageal ulcers from the medical history and confirmed with upper gastrointestinal endoscopy. The exclusion criteria were patients aged >18 years, infectious esophagitis, corrosive esophagitis caused by a detergent or a button battery, malignancy with the esophageal ulcer, connective tissue disorders manifested in the esophagus, Crohn disease manifested in the esophagus, esophageal ulcer without oral medication history, reflux symptoms persisting for >2 weeks before the drug assumption, and the use of drugs that decrease lower esophageal sphincter tone. Thirty-six patients were excluded because of esophageal ulcer without oral medication history, and two patients were excluded because their age was >18 years.

After excluding 38 patients, 32 patients were enrolled. Patient data recorded were age, sex, body height, body weight, duration of symptoms from onset to the hospital visit, causative drugs, clinical symptoms, upper gastrointestinal endoscopy findings, treatment, symptom-free duration, and complications.

Upper gastrointestinal endoscopy was performed under general anesthesia in young patients who could not cooperate, and the anesthesiologist cooperated well with us. One patient was subjected to endoscopy under general anesthesia. The endoscopy was performed in the waking state in most patients in this study, and 10% xylocaine spray was administered for local anesthesia in 31 patients before performing endoscopy. The medical service is conveniently available in our country, and we could perform endoscopy within 1 day of arranging this examination.

The Research Ethics Committee, China Medical University & Hospital, Taichung, Taiwan, approved the study with the code number CMUH110-REC1-089.

## 3. Results

The patients were 18 boys (56.2%) and 14 girls (43.8%) with a mean age of 15.6 years (range 10–18); the median age was 16 years (interquartile range 2). The mean body mass index (BMI) was 20.4 kg/m^2^ (range 14.5–25.7 kg/m^2^). The mean duration of symptoms from onset to hospital visit was 6.1 days (range 2–30). The mean duration from symptom onset to endoscopy was 8 days.

Retrosternal pain was the most common complaint (84.4%), and odynophagia was the second (43.8%) (Table 1). Doxycycline was the most common causal drug (56.3%), and NSAIDs (nonsteroidal anti-inflammatory drugs) were second (18.8%) (Table 2). The drug formulation of all the recorded pathogenic drugs in this study was capsules. Doxycycline-induced esophagitis was more frequent in boys (55.6%) than in girls (44.4%). Other patients had histories of intake of drugs such as amoxicillin, dicloxacillin, L-arginine, and clindamycin.

The number of ulcers and type were variable. They were single or multiple, mostly facing each other with apparently normal surrounding mucosa (Figure 1). There were 23 patients (71.9%) with multiple ulcers, and most had kissing ulcers (17 patients). Among the patients with kissing ulcers, 13 (76.5%) were caused by doxycycline (Figure 2). The mean number of ulcers was 2.8. Doxycycline was more frequently associated with kissing ulcers than were other drugs (72.2%, *p*-value = 0.013).

There were 25 patients (78.1%) with a lesion over the middle-third of the esophagus (Table 3). One patient had ulcers over the upper- and middle-third portion of the esophagus, and four had ulcers over the middle- to lower-third portion. Most patients (27, 84.4%) had ulcers restricted to only one part of the esophagus. There were seven patients (21.9%) with a drug-induced esophageal ulcer combined with gastric or duodenal ulcers. These ulcers were more often related to NSAIDs than other drugs (25% vs. 11.5%). The symptoms of patients with only esophageal ulcers experienced relief sooner than those with gastric or duodenal ulcers (8.7 vs. 11 days).

Thirteen patients (40.6%) were treated with sucralfate, 12 (37.5%) with PPI (proton pump inhibitors), 4 (12.5%) with H2-blockers, and 3 (9.4%) with PPI combined with sucralfate. The mean duration free from symptoms was 9.2 days (4–18 days). The duration free from symptoms was 8.36 days in the PPI treatment group, 9.38 days in the sucralfate group, and 11 days in the H2-blocker group. There was no significant difference in duration free from symptoms among PPI, sucralfate, and H2-blockers (*p*-value of PPI and H2 blocker groups was 0.15; *p*-value of the PPI and sucralfate groups was 0.70; the *p*-value of the H2 blocker and sucralfate groups was 0.46).

We scheduled a follow-up upper gastrointestinal endoscopy for 24 patients 2 months later. All 24 patients with upper gastrointestinal endoscopy had normal esophageal mucosa and no esophageal stenosis. Of the 32 patients, two were lost to follow-up after 1 week of medication, and six patients did not agree to the follow-up upper gastrointestinal endoscopy owing to symptom-free status.

## 4. Discussion

There were more male patients than female patients (56.2% and 43.8%, respectively) in this study. These results were different from those of a previous study in another country. Bordea et al. reported a predominantly female population (61.5%) and Dağ et al. also had more female patients (70.8%). Previous data from Asia published by Kim et al. showed a predominance of female patients (64.1%) [2,4,6].

Doxycycline is a common drug for treating acne [7,8]. Hanisah et al. reported that acne was more common among boys and girls with a higher prevalence of severe acne [9]. Kilkenny et al. also reported that more boys in the 15–19-year age group than girls in the same age group had acne [10]. This condition can explain why more male patients in this study had doxycycline-induced esophagitis than females. Adolescents often have acne and care about their appearance. Thus, the median age of this study was 16 years.

Moreover, the drug formulation of all recorded pathogenic drugs in this study was capsules. The capsule adheres more easily to the mucosa, and adolescents usually take capsules or tablets. However, younger children usually take medicines as powder or syrup. Powder and syrup are less likely to adhere to the esophagus mucosa. In this study, we also had a limitation that some younger-age patients had clinically suspicious symptoms but their parents did not agree to perform endoscopy. Therefore, we could not confirm the diagnosis. This also influenced the median age of this study.

The mean BMI was 20.42 kg/m^2^ in our study and was in the normal range of the age group in Taiwan [11].

Bordea et al.’s study reported an analysis of drug-induced esophagitis in children (26 patients, mean age 10.8 years). Retrosternal pain was the most common complaint (42%), and dysphagia was the second most common symptom (25%) [4]. Kim et al. reported that the clinical symptoms in adults (78 patients, mean age 43.9 years) were chest pain (71.8%), odynophagia (38.5%), and dysphagia (29.5%) [6]. These findings are similar to those noted in our study. Retrosternal pain was the most common clinical symptom for children and adolescents with drug-induced esophagitis.

Drug-induced esophageal injury tends to occur at the anatomical site of narrowing, predominantly in the middle-third portion behind the left atrium, at the level of the aortic arch, occasionally in the lower-third portion of the esophagus, and very rarely in the upper-third portion of the esophagus [3,4]. In this study, 25 patients (78.13%) had an ulcer in the middle-third portion of the esophagus. This location explained why most patients complain of retrosternal pain.

To date, several drugs have been reported to induce esophageal disorders. Antibacterial agents such as doxycycline, tetracycline, and clindamycin are offending agents in >50% of cases [12]. This finding was compatible with the findings of our study. We found that doxycycline accounted for 59.4% of the drugs in our patients, and NSAIDs accounted for 18.8%. Because doxycycline is usually prescribed for treating acne, this clarified the major cause of drug-induced esophagitis in our study [7,8]. NSAIDs are often used as antipyretic drugs in pediatric patients.

The pathophysiology of direct esophageal mucosal injury caused by drugs can be separated into transient esophagitis (self-limiting) and persistent esophagitis. Transient esophagitis is caused by doxycycline, tetracycline, clindamycin, emepronium bromide, bisphosphonates, and ferrous sulfate. When dissolved in solution, these medications have a low acid pH, causing localized discrete ulcers that heal after withdrawal of the drug and are not associated with stricture formation [3]. When held within the esophagus, doxycycline accumulates in the basal layer of squamous epithelium, a possible mechanism for tissue damage produced by this or other pills [13]. Persistent esophagitis is caused by NSAIDs, quinidine, potassium chloride, and L-arginine. NSAIDs exert their ulcerogenic effect by reducing the cytoprotective action of prostaglandins on the gastric mucosa. A similar action in the esophagus might contribute to mucosal damage and esophagitis. Long-term use (more than 10 years) of NSAIDs increases the risk of esophagitis with a stricture [3]. NSAIDs may aggravate reflux esophagitis and increase the risk of stricture formation. Arginine is an essential amino acid in juvenile humans. Arginine may promote the secretion of insulin-like growth factor 1, stimulate the immune system, and boost nitric oxide production. In our study, the patient with L-arginine-induced esophageal ulcer took L-arginine capsules to promote body growth and height. L-arginine is an L-alpha-amino acid that is the L-isomer of arginine. It is strongly alkaline (pH 10.5–12) when dissolved in water. If arginine capsules adhere to the esophageal mucosa, a local caustic injury could occur.

Many drugs that cause esophageal ulcers have been reported in the literature. Kadayifci et al. reported drug-induced esophageal ulcers caused by doxycycline, dicloxacillin, clindamycin, or amoxicillin-clavulanic acid [13,14,15,16]. Gallego Pérez et al. reported L-arginine-induced esophagitis [17,18,19].

Bordea et al. reported that 30.8% of patients had esophageal stenosis due to NSAIDs in their study. Bonavina et al. retrospectively reviewed 55 patients with a benign esophageal stricture. In 11 patients (20%), it was caused by drug-induced esophagitis due to potassium chloride, tetracyclines, aspirin, vitamin C, phenytoin, and quinidine [4,20].

We did not find any complications in our patients following endoscopy. Most cases (23, 71.9%) in our study were caused by drugs classified as transient esophagitis. Although seven cases (21.9%) were related to NSAIDs or L-arginine, i.e., the persistent esophagitis group, exposure was not long. Thus, no esophageal stricture occurred. Because drug-induced esophagitis is a chemical damage to esophageal mucosa, we must be aware of the symptoms of esophageal stricture in patients with previous drug-induced esophagitis. Most cases of drug-induced esophageal injury heal without intervention within a few days. Thus, the most important aspect of therapy is a correct diagnosis and avoiding reinjury with the drug [3].

Patients with drug-induced esophagitis often have a history of taking drugs with a small amount of water or lying down to sleep just after taking them. We strongly recommend informing patients to take medicine with enough water (approximately 100 mL) and enough time (15–30 min) before recumbency, especially high-risk drugs such as antibiotics or NSAIDs [12].

## 5. Conclusions

Physicians must be aware of the diagnosis of drug-induced esophageal ulcers in pediatric patients, especially adolescents who complain of retrosternal pain, odynophagia, or dysphagia. However, the prevalence of macrolide-resistant *Mycoplasma pneumoniae* in a tertiary care children’s center in northern Taiwan is 12.3%, and this prevalence has gradually increased. Doxycycline is used more frequently in macrolide-resistant patients [21,22]. This may be the cause of the increased prevalence of drug-induced esophagitis in young patients in the future. Moreover, some patients take L-arginine capsules to promote the growth of body height without prescription. A detailed medical history should be obtained in pediatric patients. If a patient has taken a high-risk drug that causes an esophageal ulcer, we must educate the patient about the correct method of taking medication to prevent drug-induced esophageal ulcers.

## Figures and Tables

**Figure 1 medicina-57-01286-f001:**
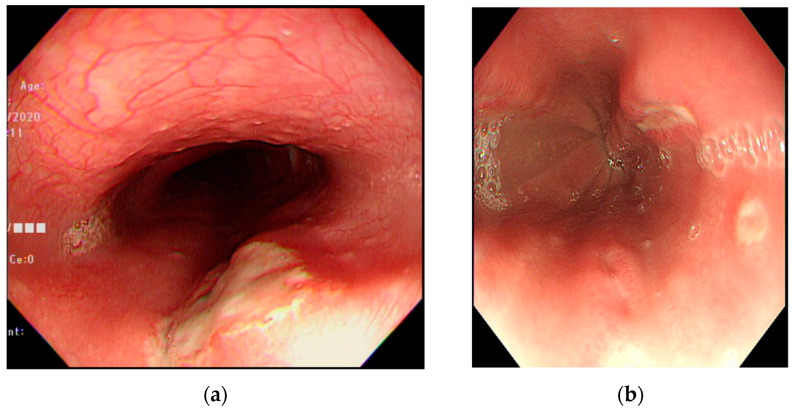
Endoscopic features of drug-induced esophageal ulcers. (**a**) A huge ulcer over the upper-third of the esophagus caused by nonsteroidal anti-inflammatory drugs in a teenage male patient. (**b**) Multiple ulcers over the middle-third of the esophagus caused by L-Arginine in a teenage female patient.

**Figure 2 medicina-57-01286-f002:**
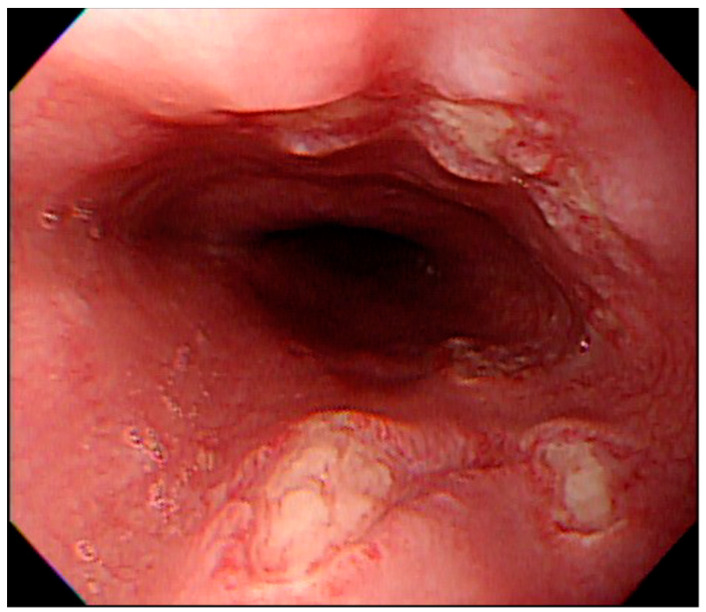
Endoscopic features of kissing ulcers. Multiple ulcers over the middle-third of the esophagus caused by doxycycline in a teenage female patient. Some of the ulcers were paired and faced each other.

**Table 1 medicina-57-01286-t001:** Demographic features and clinical symptoms of patients.

Characteristic	Value
Age (years), mean	15.6
Sex (N), Male/Female	18/14
Symptoms [N (%)]	
Chest pain	27 (84.4)
Odynophagia	14 (43.8)
Epigastric pain	12 (37.5)
Dysphagia	5 (15.6)
Duration from onset to visit (days)	6.1
Duration from onset to endoscopy (days)	8.0

**Table 2 medicina-57-01286-t002:** Pathogenic drugs identified in patients diagnosed with drug-induced esophageal ulcers.

Drug	N (%)
Doxycycline	18 (56.3)
Nonsteroidal anti-inflammatory drugs	6 (18.8)
Dicloxacillin	3 (9.4)
Amoxicillin	1 (3.1)
Clindamycin	1 (3.1)
L-Arginine	1 (3.1)
Unknown	2 (6.2)

**Table 3 medicina-57-01286-t003:** Endoscopic features of ulcers.

Features		N (%)
Location	Upper-third	5 (15.6)
	Middle-third	25 (78.1)
	Lower-third	8 (25.0)
Ulcer characteristics	Single	9 (28.1)
	Multiple	23 (71.9)
	Kissing ulcers	17 (53.1)

## Data Availability

The data that support the findings of this study are available on request from the corresponding author, S.-F.W. The data are not publicly available due to their containing information that could compromise the privacy of research participants.

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
