# Peer review of "Drug-Induced Esophageal Ulcer in Adolescent Population: Experience at a Single Medical Center in Central Taiwan"

_medicina, 2021, doi:10.3390/medicina57121286_

Round 1

Reviewer 1 Report

The authors present their results of an observational study focused on pill induced esophageal ulcers in a pediatric population. 

I find several major issues with the methodology of the study, which make interpreting the results presented by the authors very difficult:

  1. it is unclear how the study group was selected - clinical suspicion for esophageal ulcer as a trigger for endoscopy is a rather vague trigger for endoscopy, especially in a retrospective analysis.
  2. furthermore, while the authors state that they identified 70 patients with esophageal ulcer but proceeded to exclude 38, presumably (it is unclear in the manuscript!) because they met some exclusion criteria. However, some exclusion criteria should be reexamined - for instance presence of reflux symptoms or medication that could lower the pressure of the LES - since it is clearly the case that patients with reflux disease might be more prone to develop pill induced ulcers. in fact, NSAIDs are discussed by the authors themselves as being involved in a multifactorial way in the pathology of GI disease and symptoms. 
  3. the age distribution of the patient population is very important - it seems to me more like a young adult population rather than a pediatric population, with a median age of 16 ! similar studies, cited by the authors, usually have significantly lower median ages - see reference nr 4 (Bordea et al) - mean age 10.8 yrs as a an example. In this case, the authors should really focus on this aspect and discuss it as a potential limitation of their study and offer an explanation. 
  4. a clear picture of how endoscopy is performed in the authors' service is required, with high variations in practice of endoscopy, especially in pediatric populations, being well known due to local practice, custom and cultural reasons. To understand the relevance and generalizability of the results, we need to have better context for the study population

Reviewer 2 Report

THE AUTHORS HAVE CORRECTLY ANSWERED THE QUESTIONS SUBMITTED EVEN IF
THEY DID NOT DEEPEN THEM.

Author Response

This manuscript is a resubmission of an earlier submission. The following is a list of the peer review reports and author responses from that submission.

Round 1

Reviewer 1 Report

The study by Hu et al addresses an interesting topic - namely drug induced esophageal injury in a pediatric population. There is currently limited data regarding diagnostic and therapeutic endoscopic procedures in children; as such, the effort by the authors is commendable.

There are, however, several major issues which need to be addressed:

  1. the retrospective analysis design is problematic, especially when evaluating drug intake (according to the methods section) - which makes selection bias highly likely for this population. furthermore, the authors state that esophageal ulcers were diagnosed by "history taking" and further confirmed on upper GI. One has to wonder whether a large cohort of patients might not have been underdiagnosed in this way
  2. with half of the patients aged 16 or more, I wonder whether this is not more of a "young adult" population rather than a reflection of pediatric practice per se. this should be discussed in the discussion section
  3. the authors need to clearly describe the practice of endoscopy in pediatric patients to offer adequate context for their data. for instance - are very young children more likely / less likely to be offered endoscopy, since it is clear that a pediatric population is very inhomogeneous with regard to such invasive interventions, requiring sedation etc. This is a highly relevant aspect, and another important source of bias
  4. Finally, English language should be very thoroughly revised, as there are many paragraphs which are quite difficult to understand.

Reviewer 2 Report

The article is usefull because drug esophageal injury is not enough investigated in pediatric population with respect to adult population.  The article is well written and organized however the following points could be improved:

1) the follow-up period seems too short (strictures typically occur 1 to 3 months after the injury but may not develop until longer period);

2)  concomitant reflux disease patient (i.e. esophageal reflux symptoms that were persistent for greater than 2 weeks before the drug assumption) could be contemplated into the exclusion criteria as well as concomitant intake of drugs that decrease the tone of lower esophageal sphincter (i.e. anticholinergics, benzodiazepines, calcium channel blockers);

3) drug formulation could be specified (capsule or gelatin, retard formulation, etc) because higher adesivity causes more injury and this aspect should be correlated with the type of lesions.

Finally the following typing errors should be corrected:

line 165 change mort than  into more than

line 174 put the reference as kadayifes A et al

line 177 put the reference as Gallego Pérez B et al
